# Association between urinary *N*-acetyl-β-glucosaminidase activity–urinary creatinine concentration ratio and risk of disability and all-cause mortality

Shin-ichiro Tanaka[1]*, Yoshio Fujioka[2], Takeshi Tsujino[3], Tatsuro Ishida[4], Ken-ichi Hirata[4]

1 Department of Internal Medicine, Toyooka Hospital Hidaka Medical Center, Toyooka, Hyogo, Japan,
2 Division of Clinical Nutrition, Faculty of Nutrition, Kobe Gakuin University, Kobe, Japan, 3 Department of Pharmacy, School of Pharmacy, Hyogo University of Health Sciences, Kobe, Japan, 4 Division of Cardiovascular Medicine, Kobe University Graduate School of Medicine, Kobe, Japan

* shinichirou-tanaka@toyookahp-kumiai.or.jp

**Data Availability Statement:** All relevant data are within the Supporting Information files.

## Abstract

### Background

Recent studies have suggested that chronic kidney disease is associated with cardiovascular disease, dementia, and frailty, all of which cause disability and early death. We investigated whether increased activity of urinary *N*-acetyl-β-glucosaminidase (NAG), a marker of kidney injury, is associated with risk of disability or all-cause mortality in a general population.

### Methods

Follow-up data from the Hidaka Cohort Study, a population-based cohort study of members of a Japanese rural community, were obtained via questionnaires completed by participants or their relatives. Multivariable analyses were used to investigate relations between urinary NAG activity–urinary creatinine concentration ratio and risk of disability or all-cause mortality.

### Results

A total of 1182 participants were followed up for a median of 12.4 years. The endpoints were receipt of support under the public long-term care insurance program, and all-cause mortality. A total of 122 participants (10.3%) were reported to be receiving long-term care and 230 (19.5%) had died. After adjustment for cardiovascular risk factors along with physical activity, and using the quartile 1 results as a reference, the odds ratio (OR) for disability was 2.12 [95% confidence interval (95% confidence interval [CI]), 1.04–4.33; p = 0.038) and the hazard ratio (HR) for all-cause mortality was 1.65 (95% CI, 1.05–2.62; p = 0.031) in participants with urinary NAG/creatinine ratio in quartile 4. Similar results were obtained in participants

**Funding:** The author(s) received no specific funding for this work.

**Competing interests:** The authors have declared that no competing interests exist.

without proteinuria: OR for disability, 2.46 (95% CI, 1.18–5.16; p = 0.017); and HR for all-cause mortality, 1.62 (95% CI, 1.00–2.63; p = 0.049).

## Conclusions

Increased urinary NAG/creatinine ratio was associated with risk of disability or all-cause mortality in a general population.

## Introduction

Chronic kidney disease (CKD) is a growing health burden worldwide, and evidence increasingly suggests that it contributes not only to the risk of cardiovascular disease and death [1–3] but also to cognitive impairment and frailty [4–6]. Approximately 13 percent of the Japanese population aged ≥20 years has been reported to have CKD, and its prevalence increases with age [7]. Therefore, there are a considerable number of individuals with CKD among elderly Japanese. However, few studies have investigated whether decreased kidney function is independently associated with risk of disability in elderly individuals in a general population.

Values for estimated glomerular filtration rate (eGFR) represent serum creatinine–based estimates of kidney function and have been widely used in clinical and epidemiological settings. Decreased eGFR may also indicate decreased kidney function as a result of past kidney injury or simply as a result of aging. Therefore, to estimate current kidney injury, a number of biomarkers have been developed [8]. Among these, urinary *N*-acetyl-β-glucosaminidase (NAG) activity has been used to estimate tubular injury. NAG, a renal-tubular enzyme present in normal urine, is increased in a wide variety of kidney diseases [9–11]; therefore, it has for many years been regarded as a marker of renal tubular injury [8]. However, NAG activity is also closely related to development of kidney disease and enzymuria in patients with predominantly glomerular disorders [11–13]. Urinary NAG activity, reported as a normalized ratio (urinary NAG activity–urinary creatinine concentration ratio, abbreviated here as urinary NAG/creatinine ratio) to control for variation in urine flow rate, is related to urinary protein–creatinine concentration ratio [12, 14]. However, in patients with tubulointerstitial disease, enzymuria is common even in the absence of proteinuria [12]. Based on these findings, determination of urinary NAG activity may be useful for detecting both global kidney injury and non-proteinuric active kidney disease in large populations.

High urinary NAG activity has been reported to be associated with future risk of all-cause mortality and hospitalization in patients with heart failure, and measuring urinary NAG activity in addition to eGFR and urinary albumin excretion has been shown to improve prediction of these events [15]. In a population study, urinary NAG/creatinine ratio has also been shown to be associated with risk of myocardial infarction, ischemic stroke, and all-cause mortality, independently of urinary albumin excretion and cardiovascular risk factors [16]. However, in contrast to the clinical trial results [15], the NAG/creatinine ratio did not add any significant benefit to the baseline risk-prediction model. In patients with diabetes, we found conflicting results from several studies. Baseline urinary NAG activity was shown by two groups to independently predict both macro- and microalbuminuria in patients with type 1 diabetes [13, 17]; however, urinary NAG activity has been reported by another group to have no predictive value for the development of diabetic nephropathy (although this finding was based on data from only a small number of patients) [18]. Furthermore, urinary NAG activity/creatinine ratio has

been found to be associated with increased carotid intima-media thickness and carotid athero-sclerosis, independently of albuminuria, in patients with type 2 diabetes [19].

In response to the rapid aging of Japan's population, in 2000 the Japanese government began implementation, via the Long-Term Care Insurance Act [20], of a public long-term care (LTC) insurance program to deliver improved support for the country's older citizens. To test the hypothesis that urinary NAG activity may be associated with development of disability or death (from any cause) in the elderly, we used data from the Hidaka Cohort Study, a population-based study of a Japanese rural community [21–23], combined with data from the public LTC insurance program, to investigate the relation between baseline urinary NAG/creatinine ratio and risk of disability or all-cause mortality in a general Japanese population.

## Participants and methods

### Study population

The present study was a follow-up to the Hidaka Cohort Study, a population-based study of risk factors for cardiovascular disease, cancer, diabetes, and death in members of a Japanese rural community [21–23]. Data were collected from 2155 participants of that study, all of whom were aged ≥20 years at the time of the baseline survey in 1993.

The inclusion criterion for the present study was age ≥65 years at initiation of the follow-up study at the end of October 2005 and 1438 participants were eligible for this study. All participants of the age specified would have been eligible to apply for support from the Japanese government's public LTC insurance program. Recipients of support from this program are considered to have age-related impairment and were therefore defined in this study as having a disability (for details, see *Exposure and outcomes*).

Participants with a history of cardiovascular disease or cancer recorded at the time of the baseline survey were excluded from subsequent analyses. Additionally, we also excluded participants who had been reported to be bedridden or in need of nursing care at baseline (See Fig 1).

### Ethics statement

The study was carried out in accordance with the Helsinki Declaration and approved by the institutional ethical review board of Hidaka Medical Center. Informed consent was obtained from all participants or their families.

### Exposure and outcomes

The primary exposure was kidney injury, as determined by increased urinary NAG/creatinine ratio. The primary outcomes were disability (based on necessity for LTC) and all-cause mortality.

Regarding disability, under the public LTC insurance program, LTC services are provided across various settings from home- and community-based to institutional, ranging from the loan of aids for daily living and home visits from care assistants and nurses for insured persons able to live more or less independently, to day care, respite care, and continuous residential care for those with greater need for support. Thus, there are six grades of disability, and the number of available services is determined accordingly. In each case, the necessity of the requested service must be recognized by the Certification Committee for Insurance, based on their assessment of functional and cognitive status on documents, provided by family physicians, in which the extent of participants' impairments and the effects on their daily activities are described. Members of the Certification Committee also visit participants to assess their disability in accordance with uniform national standards; therefore, any person receiving a service via the public LTC insurance program is considered to have a disability; therefore, any

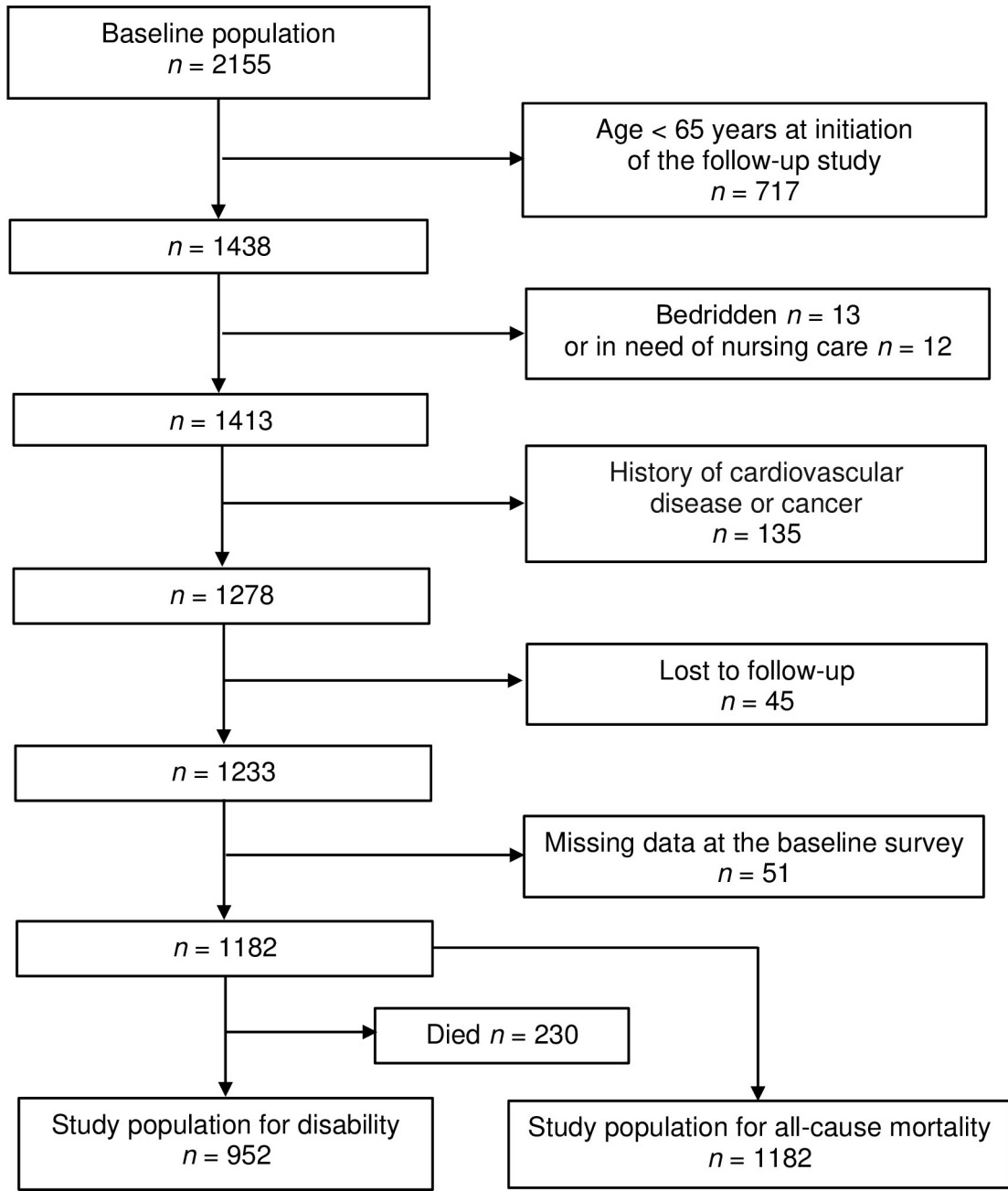

**Fig 1. Enrollment of participants in the present follow-up study of the Hidaka Cohort Study.** The study used data from those eligible for support under the Japanese government's public long-term care insurance program or who were reported to have died.

participant in the present study who was receiving LTC under the program was likewise classified as having a disability. Participants who died during the follow-up period were also regarded as having an event.

## Data collection and laboratory tests

The baseline survey was conducted between July and August 1993. The following data were collected: demographics, past medical history, history of diabetes mellitus, history of

hypertension, smoking status, whether or not the participants were bedridden, self-reported physical activity (categorized as low, moderate, or severe), body mass index, blood pressure, laboratory test results (including those for cardiovascular risk factors), and urinalysis results, as described for previous studies [21–23]. Most of the blood samples were drawn within 8 h of the participants' most recent meal; thus, the samples were obtained mainly when they were in a non-fasting state.

Urinary NAG activity was determined with the use of the synthetic substrate sodio *m*-cresolsulfonphthaleinyl (MCP) *N*-acetyl-β-D-glucosaminide [24]. MCP *N*-acetyl-β-D-glucosaminide reacts with NAG to generate MCP and *N*-acetylglucosamine. The MCP released can be determined in alkaline solution at 580 nm with a spectrophotometer by subtracting the absorbance of a MCP *N*-acetyl-β-D-glucosaminide substrate blank. The results obtained with this method correlate highly with those of the conventional fluorimetric method, which uses another substrate, 4-methylumbelliferyl *N*-acetyl-β-D-glucosaminide [24]. Spot urine samples were also used.

To avoid the dilution effects of urine samples due to variation in participants' water intake, urinary NAG activity was divided by urinary creatinine concentration to give a urinary NAG/creatinine ratio. Both serum and urinary creatinine concentration were measured by the enzymatic method.

The presence or absence of urinary protein was determined by dipstick urinalysis. Estimated glomerular filtration rate (eGFR) was calculated based on the revised equations for eGFR from serum creatinine for Japanese people [25].

## The follow-up study

In December 2005, questionnaires were mailed to participants aged ≥ 65 years to ask whether they were receiving a service under the public LTC insurance program. Relatives were asked if relevant participants had died (S1 File). Completed questionnaires were collected between January and March 2006. Therefore, the mean and median follow-up periods for total study population were 11.4 years and 12.4 years, respectively. We also asked in the follow-up study if the participants had needed nursing care at the time of the baseline survey in 1993 (S2 File).

## Statistical analyses

Continuous variables, expressed as means and standard deviations or medians and interquartile ranges, were compared by means of analysis of variance or the Kruskal–Wallis nonparametric test. Categorical variables, expressed as proportions, were compared using the chi-square test or the logistic regression model. We also used the Tukey test or Steel–Dwass test to compare data for the continuous variables between the groups. Data for urinary NAG/creatinine ratio were divided into quartiles, and risk of disability or death for quartiles 2, 3, and 4 relative to that for quartile 1 were calculated by means of the multiple logistic regression model for estimating the risk of disability and the Cox proportional hazards model for estimating all-cause mortality. $p < 0.05$ was considered to indicate a statistically significant difference.

To estimate risk of disability, we used the multiple logistic regression model rather than the Cox proportional hazards model, because data on the exact date on which participants entered the public LTC insurance program had not been collected. Furthermore, this program was implemented from April 2000 and the baseline survey was conducted in 1993; the LTC service was not available between 1993 and March 2000.

We focused on cardiovascular risk factors, because the results of recent studies have shown that most cardiovascular risk factors, such as smoking habit, hypertension, diabetes mellitus, obesity, and dyslipidemia, are also risk factors for CKD [26]. Therefore, these cardiovascular

risk factors are potential confounders for the association between kidney injury and subsequent risk of disability or all-cause mortality. Physical activity is another confounder, because participants with low daily activity could have potential disability or disease. Therefore, we added physical activity as a covariate to the multivariable model in addition to the cardiovascular risk factors. We used these risk factors in the multivariable models as covariates, including age, sex, current smoker status, body mass index, systolic blood pressure, total cholesterol, high-density lipoprotein cholesterol (HDL cholesterol), history of diabetes mellitus, and physical activity.

Urinary NAG activity has been reported to be closely associated with urinary protein concentration [12, 14]. Therefore, to investigate whether determination of urinary NAG activity would be useful for participants without proteinuria, we carried out multivariable analyses for the subgroup of participants without proteinuria. Urinary protein was recorded as a dichotomous variable. The dipstick urinalysis results *plus* or *more than plus* were recorded as 'Present', and *minus* or *plus minus* as 'Absent'.

SPSS 11.01J software for Windows (SPSS, Japan, Tokyo, Japan) and was used to perform the statistical analyses. Additionally, we used R version 4.1.2 (the R Foundation for Statistical Computing, Vienna, Austria) software for the Steel–Dwass test.

## Results

Of the 2155 participants comprising the baseline population of the Hidaka Cohort Study, 1438 participants aged $\geq$ 65 years at the initiation of the follow-up study at the end of October in 2005 were eligible for this study. A total of 1393 questionnaires were collected, giving a response rate of 96.9%. After excluding participants in bedridden (n = 13) or in need of nursing care (n = 12), with a history of cardiovascular disease or cancer (n = 135) at the time of the baseline survey, lost to follow-up (n = 45), or for whom data were missing (n = 51), data from 1182 participants were included in subsequent analyses (Fig 1). A total of 830 (70.2%) were not receiving LTC and were not reported to have died; they are subsequently reported as the no disability and alive (*NA*) group. Of the remainder, 122 (10.3%) were receiving LTC and 230 (19.5%) were reported to have died.

Table 1 summarizes the baseline characteristics of the 1182 participants whose data were included in the present study. Participants in the *Disability* or the *Had died* group were significantly older than those in the *NA* group (mean age $\pm$ standard deviation, 63.9 $\pm$ 6.3 years for *NA* group, 72.1 $\pm$6.7 years for *Disability* group, and 75.7 $\pm$ 8.4 years for *Had died* group, respectively; p < 0.001) and had decreased eGFR (p < 0.001), reflecting their higher proportion of CKD patients (p < 0.001). The considerable proportion of CKD patients in the total cohort was consistent with the proportion reported previously [7]. Notably, over forty percent of participants in the *Had died* group had CKD.

Participants in the *Had died* group were more likely to have current smoker status (p < 0.001), atrial fibrillation (p = 0.001), proteinuria (p < 0.001), and low physical activity (p < 0.001) than those in the *NA* and *Disability* groups. Other significant differences in physical activity were found between the *NA* and *Disability* groups (p = 0.001) and between the *NA* and *Had died* groups (p < 0.001). Participants in the *Disability* group were also more likely to have had a history of hypertension and had significantly higher systolic blood pressure (< 0.001 for both variables). We also found increased urinary NAG activity and urinary NAG/ creatinine ratio in both the *Disability* group and the *Had died* group (p < 0.001 for both variables).

Table 2 shows baseline characteristics and risk factors according to urinary NAG/creatinine ratio quartile. We found significantly higher values for age (p < 0.001), proportion of

**Table 1. Baseline characteristics of the participants not receiving long-term care and not reported to have died (*NA* group), participants receiving long-term care (*Disability* group), and participants reported to have died (*Had died* group)[a].**

| Characteristic | Total cohort | *NA* group | *Disability* group | *Had died* group | p[b] |
|---|---|---|---|---|---|
| No. of participants | 1182 | 830 | 122 | 230 | NA |
| Observation period, years | 12.4 (12.4–12.5) | 12.5 (12.4–12.5) | 12.5 (12.4–12.5) | 6.8 (4.4–10.1) | ND |
| Age, years | 67.1 ± 8.4 | 63.9 ± 6.3 | 72.1 ±6.7** | 75.7 ± 8.4** | < 0.001 |
| Female,% | 58.4 | 59.6 | 70.5* | 47.4** | < 0.001 |
| History of hypertension, % | 42.3 | 40.5 | 55.7** | 41.7 | 0.006 |
| History of diabetes, % | 6.3 | 6.1 | 7.4 | 6.1 | 0.865 |
| Current smoker, % | 21.5 | 19.8 | 14.8 | 31.3** | < 0.001 |
| CKD[c], % | 22.4 | 16.4 | 27.9** | 41.3** | < 0.001 |
| Presence of Af, % | 1.6 | 0.8 | 1.6 | 4.3** | 0.001 |
| Physical activity, % | | | ** [d] | ** [d] | < 0.001[b] |
| Low | 37.7 | 29.4 | 45.9 | 63.5 | NA |
| Moderate | 36.0 | 42.0 | 33.6 | 15.7 | NA |
| High | 26.2 | 28.6 | 20.5 | 20.9 | NA |
| Proteinuria, % | 4.5 | 2.3 | 8.2 | 10.4 | < 0.001 |
| Body mass index, kg/m$^2$ | 22.5 ± 3.1 | 22.9 ± 3.0 | 22.1 ± 3.0* | 21.6 ± 3.2** | < 0.001 |
| SBP, mmHg | 138 ± 22 | 137 ± 21 | 144 ± 21** | 139 ± 23 | 0.001 |
| DBP, mmHg | 78 ± 12 | 78 ± 12 | 79 ± 13 | 77 ± 12 | 0.333 |
| HbA1c, % | 5.2 (5.0–5.6) | 5.2 (5.0–5.6) | 5.2 (5.0–5.6) | 5.2 (4.9–5.6) | 0.444 |
| TC, mg/dL | 202 ± 37 | 206 ± 35 | 205 ± 40 | 188 ± 37** | < 0.001 |
| HDL cholesterol, mg/dL | 58 ± 14 | 57 ± 14 | 59 ± 14 | 59 ± 15 | 0.249 |
| Non-HDL cholesterol, mg/dL | 144 ± 39 | 148 ± 37 | 146 ± 41 | 129 ± 39** | < 0.001 |
| Triglyceride, mg/dL | 93 (69–134) | 98 (72–139) | 87 (64–123)* | 85 (63–124)** | < 0.001 |
| BUN, mg/dL | 16.1 (13.5–19.1) | 16.0 (13.4–18.6) | 16.5 (13.6–20.2) | 16.6 (13.7–19.8) | 0.054 |
| Serum creatinine, mg/dL | 0.7 (0.6–0.8) | 0.7 (0.6–0.8) | 0.7 (0,6–0.8) | 0.8 (0.7–1.0)** | < 0.001 |
| eGFR, mL/min/1.73 m$^2$ | 71 (61–78) | 73 (64–78) | 65 (58–74)** | 63 (52–73)** | < 0.001 |
| Urinary creatinine, g/L | 0.9 (0.6–1.2) | 0.9 (0.6–1.3) | 0.8 (0.6–1.1) | 0.9 (0.5–1.2) | 0.556 |
| Urinary NAG activity, U/L | 3.9 (2.5–6.3) | 3.7 (2.5–5.6) | 4.6 (3.1–7.7)** | 4.9 (2.8–8.2)** | < 0.001 |
| Urinary NAG/creatinine ratio, U/g L | 4.6 (3.3–6.7) | 4.3 (3.1–6.1) | 5.9 (4.0–7.9)** | 5.8 (4.1–8.2)** | < 0.001 |

Af, atrial fibrillation; BUN, blood urea nitrogen; CKD, chronic kidney disease; DBP, diastolic blood pressure; eGFR, estimated glomerular filtration rate; HbA1c, glycosylated hemoglobin A1c; HDL, high-density lipoprotein; NAG, *N*-acetyl-β-ᴅ-glucosaminidase; NA, not applicable; ND, not determined; SBP, systolic blood pressure; TC, total cholesterol; urinary NAG/creatinine ratio, urinary *N*-acetyl-β-ᴅ-glucosaminidase activity–urinary creatinine concentration ratio.

[a] Values expressed as mean ± standard deviation or median (interquartile range).

[b] Analysis of variance or the Kruskal–Wallis non-parametric test for continuous variables and the chi-square test for categorical variables were used to estimate p values for differences among the *NA*, *Disability*, and *Had died* groups.

[c] The percentage of participants with eGFR <60 mL/min/1.73 m$^2$.

[d] Expressing the p value for differences of low, moderate, and high physical activity categories between the *NA* and *Disability* groups or between the *NA* and *Had died* groups based on the chi-square test.

\* p < 0.05 for the difference between the *NA* and *Disability* groups or the *NA* and *Had died* groups (Tukey, Steel–Dwass test, and the chi-square test).

\*\* p < 0.01 for the difference between the *NA* and *Disability* groups or the *NA* and *Had died* groups (Tukey, Steel–Dwass test, and the chi-square test).

participants with a history of diabetes (p < 0.001), proportion of participants with proteinuria (p < 0.001), systolic blood pressure (p = 0.001), and glycosylated hemoglobinA1c (p = 0.001), and significantly lower physical activity (p = 0.001), between urinary NAG/creatinine ratio quartiles. By contrast, we found no differences among them in terms of eGFR (p = 0.584).

Table 3 shows correlations between baseline urinary NAG/creatinine ratio and eGFR and other cardiovascular risk factors and kidney injury markers. Urinary NAG/creatinine ratio

**Table 2. Baseline characteristics and risk factors according to urinary *N*-acetyl-β-ᴅ-glucosaminidase activity–urinary creatinine concentration ratio (urinary NAG/creatinine) quartile[a].**

| Baseline characteristic | Urinary NAG/creatinine quartile | | | | p[b] |
|---|---|---|---|---|---|
| | 1 | 2 | 3 | 4 | |
| No. of participants | 295 | 296 | 295 | 296 | NA |
| Urinary NAG/creatinine | 0.3–3.3 | 3.3–4.6 | 4.6–6.7 | 6.7–34.2 | NA |
| Age, years | 63.6 ± 7.2 | 66.1 ± 7.3** | 68.7 ± 8.8** | 69.8 ± 8.6** | < 0.001 |
| Female, % | 52.5 | 60.8* | 61.0* | 59.5 | 0.097 |
| History of hypertension, % | 38.6 | 39.5 | 42.0 | 49.0* | 0.047 |
| History of diabetes, % | 2.4 | 4.7 | 6.4* | 11.5** | < 0.001 |
| Current smoker, % | 19.7 | 21.3 | 22.4 | 22.6 | 0.812 |
| Presence of Af, % | 0.3 | 1.4 | 1.7 | 3.0* | 0.135 |
| Physical activity, % | | | | | |
| Low | 28.1 | 37.2 | 39.0 | 46.6 | reference |
| Moderate | 42.7 | 37.8* | 33.9** | 29.7** | < 0.001[c] |
| High | 29.2 | 25.0* | 27.1 | 23.6** | 0.010[c] |
| Proteinuria, % | 1.4 | 4.1 | 3.7 | 8.8** | 0.001 |
| Body mass index, kg/m$^2$ | 22.8 ± 2.9 | 22.3 ± 2.9 | 22.4 ± 3.3 | 22.5 ± 3.2 | 0.222 |
| SBP, mmHg | 134 ± 20 | 136 ± 22 | 138 ± 22 | 142 ± 22** | < 0.001 |
| DBP, mmHg | 78 ± 11 | 78 ± 13 | 78 ± 12 | 78 ± 12 | 0.996 |
| HbA1c, % | 5.2 (4.9–5.5) | 5.2 (5.0–5.6) | 5.2 (4.9–5.6) | 5.4 (5.0–5.8)** | 0.001 |
| TC, mg/dL | 203 ± 36 | 202 ± 35 | 203 ± 38 | 200 ± 38 | 0.572 |
| HDL-C, mg/dL | 57 ± 14 | 59 ± 14 | 58 ± 14 | 57 ± 14 | 0.068 |
| Non-HDL-C, mg/dL | 146 ± 38 | 143 ± 36 | 145 ± 40 | 143 ± 41 | 0.634 |
| Triglyceride, mg/dL | 98(60–137) | 92(61–123) | 93(64–122) | 92(61–122) | 0.133 |
| BUN, mg/dL | 16.5 (13.8–19.7) | 16.2 (13.8–19.1) | 15.7 (13.4–18.9) | 15.6 (13.3–18.9) | 0.151 |
| Serum creatinine, mg/dL | 0.7 (0.7–0.9) | 0.7 (0.6–0.8) | 0.7 (0.6–0.8)* | 0.7 (0.6–0.8) | 0.027 |
| eGFR, mL/min/1.73 m$^2$ | 67 (63–77) | 72 (62–77) | 72 (61–79) | 70 (58–79) | 0.584 |
| Urinary creatinine, mg/dl | 1.1(0.7–1.4) | 0.9(0.6–1.2) | 0.8(0.5–1.1) | 0.8(0.5–1.1) | ND |
| Urinary NAG activity, U/L | 2.6(1.7–3.5) | 2.9(0.8–3.6) | 4.5(2.9–6.2) | 7.5(4.5–10.5) | ND |

Af, atrial fibrillation; BUN, blood urea nitrogen; DBP, diastolic blood pressure; eGFR, estimated glomerular filtration rate; HbA1c, glycosylated hemoglobin A1c; HDL-C, high-density lipoprotein-cholesterol; NAG, *N*-acetyl-β-ᴅ-glucosaminidase; NA, not applicable; SBP, systolic blood pressure; TC, total cholesterol.

[a] Values expressed as mean ± standard deviation or median (interquartile range).

[b] Analysis of variance or the Kruskal–Wallis non-parametric test for continuous variables and the logistic regression model for categorical variables were used to estimate p values for differences among quartiles.

[c] p values for moderate physical activity quartiles or high physical activity quartiles compared with low physical activity quartiles were estimated using the logistic regression model.

* p < 0.05 for the difference between the Q2, Q3, or Q4 group versus the Q1 group (Tukey or Steel–Dwass test, and the logistic regression model).

** p < 0.01 for the difference between the Q2, Q3, or Q4 group versus the Q1 group (Tukey or Steel–Dwass test, and the logistic regression model).

correlated significantly with age, systolic blood pressure, and glycosylated hemoglobin A1c, suggesting a high degree of correlation between it and traditional cardiovascular risk factors. eGFR correlated significantly with age, systolic blood pressure, blood urea nitrogen, urinary creatinine, and urinary NAG activity.

Table 4 summarizes the results of univariate and multivariable-adjusted analyses for subsequent risk of disability. The results of univariate analyses with continuous variables showed this risk to be significantly higher in participants with higher NAG/creatinine ratio (odds ratio [OR], 1.15; 95% confidence interval [CI], 1.09–1.21; p < 0.001). We also found a significant increased risk of disability in participants with proteinuria than in those without proteinuria

**Table 3. Correlations between baseline urinary _N_-acetyl-β-ᴅ-glucosaminidase activity–urinary creatinine concentration ratio (urinary NAG/creatinine ratio), estimated glomerular filtration rate (eGFR), and other variables.**

| Variable | Urinary NAG/creatinine ratio | | eGFR | |
|---|---|---|---|---|
| | R | p | R | P |
| Age | 0.31 | <0.001* | −0.39 | <0.001* |
| Body mass index | −0.03 | 0.357 | −0.02 | 0.539 |
| SBP | 0.11 | <0.001* | −0.09 | 0.003* |
| DBP | −0.01 | 0.760 | 0.00 | 0.913 |
| HbA1c | 0.12 | <0.001* | 0.02 | 0.574 |
| TC | −0.03 | 0.269 | −0.01 | 0.769 |
| HDL cholesterol | −0.01 | 0.700 | 0.07 | 0.011* |
| Non-HDL cholesterol | −0.03 | 0.369 | −0.04 | 0.144 |
| Triglyceride | −0.07 | 0.016* | −0.05 | 0.118 |
| BUN | −0.06 | 0.041* | −0.28 | <0.001* |
| Serum creatinine | −0.05 | 0.097 | −0.75 | <0.001* |
| eGFR | −0.03 | 0.352 | 1.00 | NA |
| Urinary creatinine | −0.17 | <0.001* | −0.11 | <0.001* |
| Urinary NAG activity | 0.58 | <0.001* | −0.11 | <0.001* |
| Urinary NAG/creatinine ratio | 1.00 | NA | −0.03 | 0.352 |

BUN, blood urea nitrogen; DBP, diastolic blood pressure; HbA1c, glycosylated hemoglobin A1c; HDL, high-density lipoprotein; NAG, _N_-acetyl-β-ᴅ-glucosaminidase; SBP, systolic blood pressure; TC, total cholesterol.

* $p < 0.05$.

(OR, 3.81; 95% CI, 1.73–8.40; p < 0.001). Also using the results for quartile 1 as a reference, we found a significantly lower risk in participants with eGFR in quartile 3 than in those with eGFR in quartile 1 (OR, 0.28; 95% CI, 0.16–0.51; p <0.001). The ORs of urinary NAG/creatinine ratio in quartiles 3 and 4 were significantly higher: 3.29 (95% CI, 1.70–6.34; p <0.001) and 4.48 (95% CI, 2.36–8.51; p < 0.001), respectively.

In multivariable-adjusted model 1, which included the traditional cardiovascular risk factors including age, sex, systolic blood pressure, current smoker status, body mass index, total cholesterol, high-density lipoprotein-cholesterol (HDL cholesterol), and history of diabetes mellitus, urinary NAG/creatinine ratio was also significantly associated with risk of disability (OR, 2.14; 95% CI, 1.05–4.36; p = 0.036 for quartile 4 vs quartile 1; OR, 1.10; 95% CI, 1.04–1.17; p = 0.002 when urinary NAG/creatinine ratio was treated as a continuous variable). To avoid the confounder of underlying disability at the time of the baseline survey, we added physical activity as a covariate to multivariable model 2 in addition to the traditional cardiovascular risk factors. Even after adjustment for these covariates, the association of urinary NAG/creatinine ratio with risk of disability was similar to that found under model 1 (OR, 2.12; 95% CI, 1.04–4.33; p = 0.038).

Table 5 shows univariate and multivariable-adjusted hazard ratios (HRs) for subsequent all-cause mortality by kidney injury marker. The results of univariate analysis showed all the kidney injury markers to be significantly associated with risk of all-cause mortality: HR, 3.12 (95% CI, 2.04–4.76; p <0.001) for presence of urinary protein; HR, 0.32 (95% CI, 0.22–0.47; p <0.001) for eGFR quartile 3 vs quartile 1; and HR, 3.78 (95% CI, 2.44–5.86; p <0.001) for urinary NAG/creatinine ratio quartile 4 vs quartile 1. Even after adjustment for traditional cardiovascular risk factors including age, sex, systolic blood pressure, current smoker status, body mass index, total cholesterol, HDL cholesterol, and history of diabetes mellitus, urinary NAG/creatinine ratio was significantly associated with risk of all-cause mortality. We obtained a similar result with multivariable model 2, which included physical activity as a covariate.

**Table 4. Univariate and multivariable-adjusted odds ratios (ORs)[a] for subsequent disability, by kidney injury marker[b].**

| Kidney injury marker | Univariate analysis | | Multivariable model 1[c] | | Multivariable model 2[d] | |
|---|---|---|---|---|---|---|
| | OR (95% CI) | p | OR (95% CI) | p | OR (95% CI) | p |
| Urinary NAG/creatinine ratio[e] | 1.15 (1.09–1.21) | <0.001* | 1.10 (1.04–1.17) | 0.002* | 1.10 (1.04–1.17) | 0.002* |
| eGFR[e] | 0.97 (0.96–0.99) | <0.001* | 1.00 (0.98–1.02) | 0.981 | 1.00 (0.98–1.02) | 0.964 |
| Urinary protein[f] | 3.81 (1.73–8.40) | 0.001* | 3.66 (1.48–9.04) | 0.005* | 3.65 (1.48–9.02) | 0.005* |
| eGFR quartile | | | | | | |
| 1 | 1 (reference) | | 1 (reference) | | 1 (reference) | |
| 2 | 0.49 (0.30–0.81) | 0.005* | 1.26 (0.69–2.31) | 0.449 | 1.27 (0.69–2.33) | 0.438 |
| 3 | 0.28 (0.16–0.51) | <0.001* | 0.74 (0.38–1.43) | 0.371 | 0.76 (0.39–1.46) | 0.406 |
| 4 | 0.45 (0.27–0.75) | 0.002* | 1.65 (0.87–3.12) | 0.124 | 1.70 (0.90–3.23) | 0.104 |
| Urinary NAG/creatinine ratio quartile | | | | | | |
| 1 | 1 (reference) | | 1 (reference) | | 1 (reference) | |
| 2 | 1.76 (0.87–3.58) | 0.12 | 1.27 (0.59–2.74) | 0.537 | 1.27 (0.59–2.73) | 0.544 |
| 3 | 3.29 (1.70–6.34) | <0.001* | 1.96 (0.95–4.02) | 0.067 | 1.96 (0.95–4.02) | 0.068 |
| 4 | 4.48 (2.36–8.51) | <0.001* | 2.14 (1.05–4.36) | 0.036* | 2.12 (1.04–4.33) | 0.038* |

CI, confidence interval; eGFR, estimated glomerular filtration rate; urinary NAG/creatinine ratio, urinary *N*-acetyl-β-D-glucosaminidase activity–urinary creatinine concentration ratio.

[a] Odds ratios were estimated by means of the multiple logistic regression model.

[b] Data for participants reported to have died (n = 230) were excluded from the analysis.

[c] Multivariable model 1 adjusted for traditional cardiovascular risk factors including age, sex, systolic blood pressure, current smoker status, body mass index, total cholesterol, high-density lipoprotein cholesterol, and history of diabetes mellitus.

[d] Multivariable model 2 was adjusted for physical activity in addition to the traditional cardiovascular risk factors used in multivariable model 1.

[e] Included as a continuous variable.

[f] Risk for participants with proteinuria relative to that for participants without proteinuria.

* $p < 0.05$.

High urinary NAG/creatinine ratio was therefore associated with both higher risk of disability (see Table 4) and all-cause mortality (see Table 5). Furthermore, these risks (especially that for disability) increased in a stepwise fashion in line with increased urinary NAG/creatinine ratio.

Table 6 shows multivariable-adjusted ORs for disability and HRs for all-cause mortality adjusted for cardiovascular risk factors and kidney injury markers. Among the potential risk factors, only kidney injury markers were independently associated with subsequent risk of disability. We also found current smoker status, low total cholesterol, urinary protein, and urinary NAG/creatinine ratio to be independently associated with risk of all-cause mortality. Therefore, among various risk factors, only kidney injury markers, urinary protein, and urinary NAG/creatinine ratio were significantly associated with both subsequent risk of disability and all-cause mortality.

We found presence of urinary protein to be an independent risk factor for risk of disability and all-cause mortality. Therefore, we investigated whether urinary NAG/creatinine ratio could predict subsequent risk of disability and all-cause mortality in participants *without* proteinuria. We performed a multivariable analysis adjusted for age, sex, systolic blood pressure, current smoker status, body mass index, total cholesterol, HDL cholesterol, history of diabetes mellitus, and physical activity in participants *without* proteinuria.

Table 7 shows the ORs and HRs for disability and all-cause mortality, stratified by urinary NAG/creatinine ratio quartiles in participants *without* proteinuria. The participants with urinary NAG/creatinine ratio in quartile 4 were at higher risk for disability (OR, 2.46; 95% CI,

**Table 5. Univariate and multivariable-adjusted hazard ratios (HRs)[a] for subsequent all-cause mortality, by kidney injury marker.**

| Kidney injury marker | Univariate analysis | | Multivariable model 1[b] | | Multivariable model 2[c] | |
|---|---|---|---|---|---|---|
| | HR (95% CI) | p | HR (95% CI) | p | HR (95% CI) | p |
| Urinary NAG/creatinine ratio[d] | 1.09 (1.07–1.12) | <0.001* | 1.07 (1.04–1.10) | <0.001* | 1.07 (1.03–1.10) | <0.001* |
| eGFR[d] | 0.97 (0.96–0.98) | <0.001* | 1.00 (0.99–1.01) | 0.474 | 1.00 (0.99–1.01) | 0.662 |
| Urinary protein[e] | 3.12 (2.04–4.76) | <0.001* | 2.01 (1.30–3.09) | 0.002* | 1.97 (1.28–3.03) | 0.002* |
| eGFR quartile | | | | | | |
| 1 | 1 (reference) | | 1 (reference) | | 1 (reference) | |
| 2 | 0.37 (0.26–0.53) | <0.001* | 0.77 (0.54–1.11) | 0.166 | 0.77 (0.53–1.10) | 0.152 |
| 3 | 0.32 (0.22–0.47) | <0.001* | 0.68 (0.46–1.02) | 0.059 | 0.73 (0.49–1.08) | 0.116 |
| 4 | 0.37 (0.26–0.53) | <0.001* | 1.00 (0.67–1.50) | 0.994 | 1.05 (0.70–1.58) | 0.800 |
| Urinary NAG/creatinine ratio quartile | | | | | | |
| 1 | 1 (reference) | | 1 (reference) | | 1 (reference) | |
| 2 | 1.91 (1.19–3.09) | 0.008* | 1.36 (0.84–2.22) | 0.213 | 1.34 (0.82–2.18) | 0.239 |
| 3 | 2.81 (1.79–4.42) | <0.001* | 1.34 (0.84–2.15) | 0.218 | 1.30 (0.81–2.09) | 0.271 |
| 4 | 3.78 (2.44–5.86) | <0.001* | 1.72 (1.09–2.72) | 0.019* | 1.65 (1.05–2.62) | 0.031* |

CI, confidence interval; eGFR, estimated glomerular filtration rate; urinary NAG/creatinine ratio, urinary *N*-acetyl-β-D-glucosaminidase activity–urinary creatinine concentration ratio.

[a] Estimated by means of the Cox regression model.

[b] Multivariable model 1 was adjusted for age, sex, systolic blood pressure, current smoker status, body mass index, total cholesterol, high-density lipoprotein cholesterol, and history of diabetes mellitus.

[c] Multivariable model 2 was adjusted for physical activity in addition to the variables used in multivariable model 1.

[d] Included as a continuous variable.

[e] Risk for participants with proteinuria relative to that for participants without proteinuria.

* $p < 0.05$.

1.18–5.16; p = 0.017) and all-cause mortality (HR, 1.62; 95% CI, 1.00–2.63; p = 0.049). These risks increased in a stepwise fashion in line with increased NAG/creatinine ratio.

## Discussion

In this community-based cohort study of 1182 persons who were apparently healthy at the time of the baseline survey, increased urinary NAG/creatinine ratio was found to be associated with subsequent risk of disability or all-cause mortality, independent of traditional cardiovascular risk factors including age. The risk increased in a stepwise fashion from the lowest to the highest urinary NAG/creatinine ratio quartile. This relation remained significant even in participants without proteinuria. Furthermore, among various cardiovascular risk factors, only kidney injury markers were associated with both the risk of disability and the risk of all-cause mortality in this cohort, in which over twenty percent of participants had CKD. Therefore, we should focus on CKD to reduce future risk of disability and death.

A number of studies have been carried out to investigate the relation between CKD and cognitive impairment and frailty. However, the mechanism by which CKD leads to subsequent disability is not yet fully understood. One possible mechanism may be failure of appropriate blood pressure control. A previous study revealed that even in the early stage of CKD, there are considerable increases in nocturnal blood pressure [27]. However, during the follow-up period, nocturnal hypertension had not been well-recognized by most physicians because ambulatory blood pressure monitoring had not yet been approved for use in routine clinical practice during the follow-up period. Therefore, participants with potentially decreased kidney function might not have received treatment to achieve appropriate blood pressure control. By

**Table 6. Multivariable-adjusted odds ratios (ORs) for disability and hazard ratios (HRs) for all-cause mortality adjusted for cardiovascular risk factors and kidney injury markers.**

| Variable[a] | Disability | | All-cause mortality | |
|---|---|---|---|---|
| | OR (95% CI) | p | HR (95% CI) | p |
| Male sex | 0.69 (0.37–1.28) | 0.236 | 1.34 (0.95–1.89) | 0.091 |
| Age | 1.18 (1.14–1.23) | <0.001* | 1.13 (1.11–1.16) | <0.001* |
| History of diabetes | 1.03 (0.44–2.40) | 0.944 | 1.04 (0.60–1.80) | 0.890 |
| Current smoker status | 1.23 (0.59–2.55) | 0.580 | 1.85 (1.30–2.63) | 0.001* |
| Body mass index | 0.95 (0.88–1.03) | 0.183 | 0.98 (0.94–1.03) | 0.423 |
| SBP | 1.01 (1.00–1.02) | 0.173 | 1.00 (0.99–1.00) | 0.181 |
| TC | 1.00 (0.99–1.01) | 0.729 | 0.99 (0.99–1.00) | <0.001* |
| HDL cholesterol | 1.01 (0.99–1.03) | 0.236 | 1.01 (1.00–1.02) | 0.197 |
| Physical activity | | 0.483[b] | | 0.016[b*] |
| Low | 1 (reference) | NA | 1 (reference) | NA |
| Moderate | 0.82 (0.49–1.37) | 0.440 | 0.62 (0.42–0.92) | 0.018* |
| High | 0.71 (0.39–1.28) | 0.252 | 0.66 (0.46–0.95) | 0.027* |
| eGFR | 1.00 (0.98–1.02) | 0.940 | 1.00 (0.99–1.01) | 0.949 |
| Urinary protein | 3.32 (1.32–8.34) | 0.011* | 1.74 (1.11–2.72) | 0.015* |
| Urinary NAG/creatinine | 1.10 (1.03–1.17) | 0.003* | 1.06 (1.03–1.09) | <0.001* |

CI, confidence interval; eGFR, estimated glomerular filtration rate; HDL, high-density lipoprotein; NA, not applicable; SBP, systolic blood pressure; TC, total cholesterol; urinary NAG/creatinine, urinary *N*-acetyl-β-D-glucosaminidase activity–urinary creatinine concentration ratio.

[a] All variables were included in the logistic regression model and the Cox proportional regression model to estimate the risk associated with each variable.

[b] Expressed as p value among three categories of physical activity.

*p < 0.05.

contrast, participants who were recognized as having daytime hypertension at the time of the baseline survey might have received appropriate treatment, and therefore baseline high blood pressure may no longer be associated with subsequent risk of disability and all-cause mortality.

Further support for the idea that CKD is closely related to aging is provided by the finding that patients with CKD have decreased expression of Klotho and increased fibroblast growth factor 23 levels in accordance with CKD stage (1 to 5) [28, 29]. The results of studies using animal models have suggested that Klotho, which as a coreceptor for fibroblast growth factor 23 plays a critical role in phosphate excretion by the kidney, is closely related to accelerated aging and early death [30–33]. This evidence is supported by results of human studies, which have

**Table 7. Multivariable-adjusted odds ratios (ORs)[a] and hazard ratios (HRs)[a] for subsequent disability and all-cause mortality according to urinary *N*-acetyl-β-D-glucosaminidase activity–urinary creatinine concentration ratio (urinary NAG/creatinine ratio) quartile in participants *without* proteinuria.**

| Urinary NAG/creatinine ratio quartile | Disability | | All-cause mortality | |
|---|---|---|---|---|
| | OR (95% CI) | p | HR (95% CI) | p |
| 1 | 1 (reference) | NA | 1 (reference) | NA |
| 2 | 1.09 (0.48–2.46) | 0.843 | 1.33 (0.80–2.22) | 0.274 |
| 3 | 1.99 (0.94–4.23) | 0.074 | 1.41 (0.87–2.31) | 0.164 |
| 4 | 2.46 (1.18–5.16) | 0.017* | 1.62 (1.00–2.63) | 0.049* |

CI, confidence interval; NA, not applicable.

[a] Both ORs and HRs were estimated by means of the logistic regression model and the Cox regression model adjusted for age, sex, systolic blood pressure, current smoker status, body mass index, total cholesterol, high-density lipoprotein-cholesterol, history of diabetes mellitus, and physical activity.

* p < 0.05.

shown increased blood fibroblast growth factor-23 levels to be independently associated with increased risk of vascular and nonvascular mortality and disability in a general population [34, 35]. Furthermore, low plasma Klotho concentrations have been shown to be independently associated with disability [36], cognitive decline [37], and mortality [38] in older, community-dwelling persons.

Chronic kidney disease has been defined as having decreased eGFR, therefore eGFR has been used to assess the severity of CKD [39]. However, eGFR was determined with the use of serum creatinine concentration, along with age and sex as covariates. Therefore, adjusting for age and sex in a multivariable model seems to be 'over-adjustment'. This may lead to a non-significant association between eGFR and risk of disability and all-cause mortality in multivariable models. We found additional benefit in measuring urinary NAG/creatinine ratio, because we found no differences in eGFR among the urinary NAG/creatinine ratio quartiles (see Table 2). Furthermore, even after adjustment for eGFR, urinary protein, and traditional cardiovascular risk factors, urinary NAG/creatinine ratio remained significantly associated with risk of disability and all-cause mortality. Therefore, measurement of urinary NAG/creatinine ratio may have a benefit in addition to measurement of eGFR and urinary protein.

## Limitations

This study has several limitations. First, we did not evaluate disability at the baseline survey in 1993 unless the participant was bedridden, and the Long-Term Care Insurance Act was not passed until 2000. Therefore, we had no contemporary data on the participants' disabilities at baseline. In the follow-up study, we asked participants retrospectively about their baseline disability status. However, this later information may not have been sufficient to identify participants with mild to moderate disability at the time of the baseline survey. Therefore, we added physical activity as a covariate to the multivariable models in order to adjust for this confounder.

Second, we did not collect data on participants' disability status but simply asked if they were using the public LTC insurance program. Therefore, participants with a disability but who did not use the service might not have been recognized as having a disability. This may have contributed to reducing the strength of the association between baseline urinary NAG/creatinine ratio and risk of disability in this study.

Third, although urinary protein concentration has commonly been used as a marker of kidney injury and shown to be closely related to urinary NAG/creatinine ratio [12, 14], only the presence or absence of urinary protein was determined in the present study. Therefore, it was not possible to investigate the relation between baseline urinary protein concentration and subsequent risk of disability or all-cause mortality. However, because the great majority of the total cohort did not have proteinuria, and in these participants, urinary NAG/creatinine ratio was associated with subsequent risk of disability and death, determination of urinary NAG activity may predict this risk in most members of a general population.

Fourth, urinary NAG/creatinine ratio was determined only once, at the baseline survey, and > 12 years had passed before the data for disability or all-cause mortality were collected. This may have resulted in underestimation of the relation between urinary NAG/creatinine ratio and subsequent disability or all-cause mortality [40].

## Conclusions

Increased urinary NAG/creatinine ratio was associated with both risk of subsequent disability and risk of all-cause mortality, independent of traditional cardiovascular risk factors. The results of our study support the findings of previous studies in suggesting that chronic kidney

injury is closely related to aging. For prediction of risk of disability or all-cause mortality, urinary NAG/creatinine ratio was useful even in persons without proteinuria. Therefore, urinary NAG activity may be one of the most useful markers of kidney injury in terms of predicting subsequent risk of disability or all-cause mortality in a general population. This is especially important in aging populations such as that of Japan.

## Supporting information

**S1 File. Screening questionnaire concerning long-term care service use Part 1.**
(DOC)

**S2 File. Screening questionnaire concerning long-term care service use Part 2.**
(DOCX)

**S1 Data.**
(XLSX)

## Acknowledgments

We wish to thank the study participants and their family physicians. We also greatly appreciate the technical assistance provided by the staff of Hidaka Medical Center and Kobe University.

## Author Contributions

**Conceptualization:** Shin-ichiro Tanaka.

**Data curation:** Shin-ichiro Tanaka, Yoshio Fujioka.

**Formal analysis:** Shin-ichiro Tanaka.

**Methodology:** Shin-ichiro Tanaka.

**Project administration:** Shin-ichiro Tanaka, Yoshio Fujioka.

**Supervision:** Yoshio Fujioka, Takeshi Tsujino, Tatsuro Ishida, Ken-ichi Hirata.

**Validation:** Yoshio Fujioka, Takeshi Tsujino, Tatsuro Ishida.

**Writing – original draft:** Shin-ichiro Tanaka.

**Writing – review & editing:** Yoshio Fujioka.

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
