## [Decision Letter · Decision Letter 0]

1 Nov 2021

PONE-D-21-27646Urinary N-acetyl-β-glucosaminidase activity–urinary creatinine concentration ratio predicts risk of disability or early deathPLOS ONE

Dear Dr. Tanaka,

Thank you for submitting your manuscript to PLOS ONE. After careful consideration, we feel that it has merit but does not fully meet PLOS ONE’s publication criteria as it currently stands. Therefore, we invite you to submit a revised version of the manuscript that addresses the points raised during the review process.

Three Reviewers all raised concerns about the follow-up methods and statistical analyses of the study. Please carefully consider these requests during revision.

We look forward to receiving your revised manuscript.

Kind regards,

Yan Li, MD, PhD

Academic Editor

PLOS ONE

Journal Requirements:

2. Please upload an English language copy of the questionnaire as a supplementary file.

Reviewers' comments:

Reviewer's Responses to Questions

**Comments to the Author**

1. Is the manuscript technically sound, and do the data support the conclusions?

Reviewer #1: Partly

Reviewer #2: Partly

Reviewer #3: No

2. Has the statistical analysis been performed appropriately and rigorously? 

Reviewer #1: Yes

Reviewer #2: No

Reviewer #3: No

3. Have the authors made all data underlying the findings in their manuscript fully available?

Reviewer #1: Yes

Reviewer #2: No

Reviewer #3: No

4. Is the manuscript presented in an intelligible fashion and written in standard English?

Reviewer #1: Yes

Reviewer #2: Yes

Reviewer #3: Yes

5. Review Comments to the Author

Reviewer #1: The authors evaluated whether increased urinary N-acetyl-β-glucosaminidase activity to creatinine ratio (NCR) sampled from spot urine was predictive of disability or death in a total of 1,209 elderly Japanese. The authors performed multivariable logistic regression and found that subjects with the highest quartile of NCR had increased risk of disability, death, and combination of the two.

Overall, the paper contained valid information on NCR and adverse outcome. However, there are several significant issues that need to be addressed.

Major comments

My first major concern is about the description of the study, which is different from previously published papers (reference 19-21).

1. The baseline of the current study, unlike the what the authors wrote in the abstract, the methods and the results that "this study started in December 2005", was collected at year 1993.

2. Follow-up was conducted more than ten years later, around 2005. The endpoints including disability and death were collected at this period.

The authors should 1) clarify the exact inclusion criteria, e.g., aged >= 65 years at the time of follow-up. 2) correct all related sentences and provide the duration of the follow-up. 3) conduct additional analyses including Cox regression for all-cause mortality, and if possible, for the composite outcome including all-cause mortality and disability, when exact date of disability was possible. 4) modify the study flow chart accordingly (figure 1).

Secondly, given the long-term care act was implemented around year 2000, and that the disability status could not be determined at baseline ("apparently healthy"), the conclusion that NCR PREDICTED risk of disability could not be drawn. Unless additional information was provided, baseline NCR was ASSOCIATED with disability in the cross-sectional survey conducted at follow-up. Meanwhile, were all subjects screened for eligibility of LTC or did they need to apply for it?

Thirdly, additional statistical analysis concern should be addressed apart from the Cox regression as mentioned in the first major comment.

1. The ranges for NCR quartiles, numbers and rates of events should be provided.

2. The Relative Risk should be Odds Ratio in logistic regression. A raw calculation from the data provided in the supplemental file also confirmed the need to replace the term RR with OR.

3. Analysis for continuous NCR should be conducted.

4. Re-classification of predicted risk after adding the NCR into base model should be analyzed, even if non-significant. This is particularly important given the fact that the age of subjects in DD group on average was 10 years older than NA group and reached the healthy life expectancy of 74.1 years.

Fourthly, introduction and discussion section should be revised to give readers a clear and objective view on the current finding.

1. The population included in the current study were not all CKD patients, indeed most of them were free of CKD. First paragraph of introduction focused on CKD, which is not the population, nor the disease need to diagnose in the population. The authors may, for instance, give a summary of kidney biomarkers and population risk here.

2. Details on the act of long-term care and its use in the determination of disability should be put into methods section. Disability, on the other hand, could be discussed here as a general term.

3. The conclusion in the first paragraph of discussion that subjects with proteinuria could not benefit from NCR test is underpowered given there were only 56 subjects with proteinuria. The conclusion therefore should be downplayed.

4. The comparison of current study with existing evidence on NCR and adverse outcome should be discussed firstly. Multiple studies focused on this topic could be retrieved from PubMed search, e.g., the paper by Solbu et al. published in J Am Soc Nephrol (27: 533–542, 2016, doi: 10.1681/ASN.2014100960), compared NCR with albumin-to-creatinine ratio in risk prediction of mortality in a low-risk population.

5. The relationship among NCR/CKD, aging and cardiovascular risk should be clearly illustrated. In the second paragraph of discussion the authors made a bold statement that CKD is related to physiological change of aging rather than that of CVD, which is not appropriate as in the following paragraph the authors explained mechanisms involving in such procedure, i.e., lack of BP control, whereas hypertension is one of the most important CV risk factors.

6. Is there any interplay between NCR and Klotho and FGF23? Please provide pathophysiological and/or epidemiological links here given the current study population was only general but not CKD population. Alternatively, replace them with other evidence on the pathophysiological importance of NCR.

7. Given the current guidelines from KDIGO and other organizations all recommend the use of ACR and eGFR in CKD staging, whether NCR levels were indicative of CKD stage should be addressed before further discussion on CKD.

8. eGFR is an established risk marker of CKD but it should be used in conjunction with others when eGFR is not low (<60ml/min/1.73m2). Current data suggested the population included here is not particularly of impaired renal filtration. Relevant discussion on eGFR in comparison to NCR should be shortened and organized.

Minor comments

To make it less confusing, only keep the number of included subjects in the abstract and delete the total number screened.

Line 127 of page 6, "an" MCP should be "a" MCP.

In the methods section, please specify the name and version of statistical software.

In the methods section, please specify the regression models used for the evaluation.

Table 4 has duplicated results with Table 3 (the "entire cohort"). Similarly Figure 2 is a graphical presentation of some results presented in Table 3. Such items should be removed.

Conclusion "public LTC insurance services" should be replaced by disability or disability determined by public LTC insurance services.

Reviewer #2: In this manuscript, the author has demonstrated that the activity of urinary N-acetyl-β-glucosaminidase (NAG), a marker of kidney injury, is associated with subsequent risk of disability or early death in a general population. However, there exists several problems.

1、 What is the specific date on which to determine whether the events occurred or not？Please describe the medium and IQR of the follow-up period.

2、 The association between the quartile groups of NAG groups and disability or death should be evaluated using the Kaplan-Meier survival method and compared using log-rank statistics.

3、 How about the relationships between urinary NAG and other clinical variables?

4、 The association between the NAG and cardiovascular mortality and renal mortality should better be evaluated.

5、 What is your criteria for selecting independent variables in your multivariable-adjusted model？

6、 The usage of anti-hypertensive drugs, anti-diabetic drug, statins and comorbid conditions should be recorded and adjusted.

7、 In order to determine whether risk prediction models were improved by addition of the NAG, C-index should be calculated for the demographic, eGFR, and cardiovascular risk factor model for each outcome.

8、 Q1 needs to be placed in the table, and the specific values of Q1-Q4 need to be marked.

9、 NAG is one of the urine kidney injury biomarkers, not equals to CKD. These two concepts should not be confused.

10、 In line 277-279, you mentioned that “The association between urinary NAG/creatinine ratio and disability or death, which was found to be independent of cardiovascular risk factors (including age), suggests that CKD is related to the physiological changes of aging rather than those of cardiovascular disease. ”, while I am so confused about this sentence. How can I come to a conclusion that “CKD is related to the physiological changes of aging” from your data? The discussion should be re-arranged.

11、 In your opinion, what is the probably reason that urinary NAG/creatinine ratio can not predict subsequent disability or death in patients with proteinuria?

12、 I think Table 2 and Figure 2 maybe not necessary.

Reviewer #3: The description of the study design in this article is incomplete and lacks specific follow-up methods. There are also doubts about the statistical methods used in this article to illustrate the relationship between urinary NAG/creatinine ratio and the risk of death and disability. So I don't think this article has reached the standard for receiving manuscripts. Questions are as follows:

1. What is the collection method of the outcome event data? At what time were the collection points?

2. Since the beginning of the study in 2005, what was the median follow-up time for all participants? How many subjects were lost to follow-up?

3. The statistical method used in the analysis of survival data in this paper used multivariate logistic regression analysis instead of the Cox risk regression model (Table 2 and Table 3), which ignored the impact of censored data.

4. In Table 4, participants with proteinuria were only 56, which could produce an unstable model.

6. PLOS authors have the option to publish the peer review history of their article (what does this mean?). If published, this will include your full peer review and any attached files.

Reviewer #1: No

Reviewer #2: No

Reviewer #3: No

---

## [Author Response · Author response to Decision Letter 0]

29 Dec 2021

Reviewer #1: The authors evaluated whether increased urinary N-acetyl-β-glucosaminidase activity to creatinine ratio (NCR) sampled from spot urine was predictive of disability or death in a total of 1,209 elderly Japanese. The authors performed multivariable logistic regression and found that subjects with the highest quartile of NCR had increased risk of disability, death, and combination of the two.

Overall, the paper contained valid information on NCR and adverse outcome. However, there are several significant issues that need to be addressed.

Major comments

My first major concern is about the description of the study, which is different from previously published papers (reference 19-21).

1. The baseline of the current study, unlike the what the authors wrote in the abstract, the methods and the results that "this study started in December 2005", was collected at year 1993.

Response

Thank you for pointing this out. We have specified the observation period in the results section of the abstract as follows: ‘A total of 1182 participants were followed up for a median of 12.4 years’ (line 30).

2. Follow-up was conducted more than ten years later, around 2005. The endpoints including disability and death were collected at this period.

The authors should 1) clarify the exact inclusion criteria, e.g., aged >= 65 years at the time of follow-up. 2) correct all related sentences and provide the duration of the follow-up. 3) conduct composite outcome including all-cause mortality and disability, when exact date of disability was possible. 4) modify the study flow chart accordingly (figure 1).

Response

1. The inclusion criterion is now clarified in the Study population subsection of Participants and Methods: ‘The inclusion criterion for the present study was age >65 years at initiation of the follow-up study at the end of October 2005’ (lines 96–97).

2. The dates of the baseline survey and follow-up study are now specified in the Data collection and laboratory tests and The follow-up study subsections, respectively, of Participants and Methods: ‘The baseline survey was conducted between July and August 1993’ (line 130) and ‘Completed questionnaires were collected between January and March 2006’ (line 155-156). The follow-up periods are also specified: ‘Therefore, the mean and median follow-up periods for total study population were 11.4 years and 12.4 years, respectively’ (lines 156–157).

3. Participants were not asked their exact date of entry into the LTC insurance program. Therefore, the Cox proportional hazards regression model could not be used to estimate the risk of disability. Instead, we conducted an additional analysis using another Cox regression model to investigate the relation between baseline urinary NAG/creatinine ratio and risk of all-cause mortality. This is explained in the Statistical analyses subsection thus: ‘To estimate risk of disability, we used the multiple logistic regression model rather than the Cox proportional hazards model, because data on the exact date on which participants entered the public LTC insurance program had not been collected.’ (lines 169-171).

4. The participant selection flow chart (Fig 1) has been modified accordingly.

Secondly, given the long-term care act was implemented around year 2000, and that the disability status could not be determined at baseline ("apparently healthy"), the conclusion that NCR PREDICTED risk of disability could not be drawn. Unless additional information was provided, baseline NCR was ASSOCIATED with disability in the cross-sectional survey conducted at follow-up. Meanwhile, were all subjects screened for eligibility of LTC or did they need to apply for it?

Response

We agree with your point and have therefore added information to the analysis. At the baseline survey, we identified participants who were bedridden, and in the follow-up study, we asked participants if they had needed nursing care at the time of the baseline study. Participants who were bedridden or needed nursing care at baseline have been excluded from the analysis. Additionally, we have added baseline physical activity to the model as a covariate, to adjust for potentially underlying disability and disease in participants with low physical activity (‘Physical activity is another confounder, because participants with low daily activity could have potential disability or disease...’; lines 178–180).

As explained in the Limitations subsection of the Discussion, ‘we did not collect data on participants’ disability status but simply asked if they were using the public LTC insurance program’ (lines 422–423); their disability status was confirmed by officials who visited them at home. ‘Therefore, participants with a disability but who did not use the service might not have been recognized as having a disability. This may have contributed to reducing the strength of the association between baseline urinary NAG/creatinine ratio and risk of disability in this study.’ (lines 423–426). It is possible that such participants did not need any government support.

Thirdly, additional statistical analysis concern should be addressed apart from the Cox regression as mentioned in the first major comment.

1. The ranges for NCR quartiles, numbers and rates of events should be provided.

Response 

We have added baseline characteristic data for urinary NAG–creatinine ratio quartiles to Table 2. We have also presented the numbers of events in Table 1 and the Results (‘A total of 830 (70.2%) were not receiving LTC and were not reported to have died; they are subsequently reported as the no disability and alive (NA) group. Of the remainder, 122 (10.3%) were receiving LTC and 230 (19.5%) were reported to have died.’; lines 200–202).

2. The Relative Risk should be Odds Ratio in logistic regression. A raw calculation from the data provided in the supplemental file also confirmed the need to replace the term RR with OR.

Response 

Thank you for this information, which is important for the presentation of the data. We have changed the relevant terms as advised.

3. Analysis for continuous NCR should be conducted.

Response 

We have added the results of multivariable adjusted analyses for continuous NCR to Tables 4, 5, and 6. In each case, NCR was significantly associated with both outcomes.

4. Re-classification of predicted risk after adding the NCR into base model should be analyzed, even if non-significant. This is particularly important given the fact that the age of subjects in DD group on average was 10 years older than NA group and reached the healthy life expectancy of 74.1 years.

Response

Thank you for your important point about the analyses.

We have included all variables in the multivariable-adjusted models that we used in the current study, and we have described every predicted risk for each variable in Table 6.

Fourthly, introduction and discussion section should be revised to give readers a clear and objective view on the current finding.

1. The population included in the current study were not all CKD patients, indeed most of them were free of CKD. First paragraph of introduction focused on CKD, which is not the population, nor the disease need to diagnose in the population. The authors may, for instance, give a summary of kidney biomarkers and population risk here.

Response 

Thank you for your helpful comment.

In the Introduction, we refer to the prevalence of CKD in the Japanese population (13% of persons aged >20 years). Additionally, we give the prevalence of CKD (defined as eGFR <60 mL/min/1.73 m2) in our cohort: 22.4% (see Table 1). We also refer to previous reports that high urinary NAG activity is associated with the risk of cardiovascular disease, heart failure, and mortality (‘High urinary NAG activity has been reported to be associated with future risk of all-cause mortality and hospitalization in patients with heart failure, and measuring urinary NAG activity in addition to eGFR and urinary albumin excretion has been shown to improve prediction of these events [15]. In a population study, urinary NAG/creatinine ratio has also been shown to be associated with risk of myocardial infarction, ischemic stroke, and all-cause mortality, independently of urinary albumin excretion and cardiovascular risk factors [16].’; lines 67–72).

2. Details on the act of long-term care and its use in the determination of disability should be put into methods section. Disability, on the other hand, could be discussed here as a general term.

Response 

Thank you for this helpful comment. In the Participants and methods section, we now state how we have defined disability in the Study population subsection (‘Recipients of support from this program are considered to have age-related impairment and were therefore defined in this study as having a disability’; lines 99–101) and provide details of long-term care in the Exposure and outcomes subsection (lines 114–126).

3. The conclusion in the first paragraph of discussion that subjects with proteinuria could not benefit from NCR test is underpowered given there were only 56 subjects with proteinuria. The conclusion therefore should be downplayed.

Response 

In accordance with your recommendation, we have downplayed this statement in the first paragraph of the Discussion (‘This relation remained significant even in participants without proteinuria.’; lines 375–376).

4. The comparison of current study with existing evidence on NCR and adverse outcome should be discussed firstly. Multiple studies focused on this topic could be retrieved from PubMed search, e.g., the paper by Solbu et al. published in J Am Soc Nephrol (27: 533–542, 2016, doi: 10.1681/ASN.2014100960), compared NCR with albumin-to-creatinine ratio in risk prediction of mortality in a low-risk population.

Response 

In the Introduction, we refer to previous reports that focused on the relation between NCR and risk of cardiovascular diseases and mortality (lines 67–80). However, we are willing to move this information to the Discussion, if advised to do so.

5. The relationship among NCR/CKD, aging and cardiovascular risk should be clearly illustrated. In the second paragraph of discussion the authors made a bold statement that CKD is related to physiological change of aging rather than that of CVD, which is not appropriate as in the following paragraph the authors explained mechanisms involving in such procedure, i.e., lack of BP control, whereas hypertension is one of the most important CV risk factors.

Response 

We refer to nocturnal hypertension as an example of ‘masked hypertension’. People with this condition may not be recognized as having hypertension in epidemiological studies based on office hypertension, leading to misclassification of the disease (lines 382–388).

We did not perform ambulatory blood pressure monitoring in this study, so we were unable to confirm this hypothesis.

6. Is there any interplay between NCR and Klotho and FGF23? Please provide pathophysiological and/or epidemiological links here given the current study population was only general but not CKD population. Alternatively, replace them with other evidence on the pathophysiological importance of NCR.

Response 

NCR simply represents tubular and interstitial injury of the kidney. We were unable to find any interactions between NCR and FGF23 or Klotho. However, all of them relate to kidney function, and considerable increases in NCR and FGF23 and a decrease in Klotho in relation to decreases in kidney function have already been reported.

In addition to studies using experimental animal models, we found a number of epidemiological studies regarding relations between Klotho and FGF23 and the future risk of cardiovascular disease, death, and frailty in the general population. We have added this information to the Discussion (lines 392–401).

7. Given the current guidelines from KDIGO and other organizations all recommend the use of ACR and eGFR in CKD staging, whether NCR levels were indicative of CKD stage should be addressed before further discussion on CKD.

Response 

In the Discussion, we propose using additional kidney injury markers alongside eGFR and ACR to achieve improved estimates of future risk of disability and death in the general population.

8. eGFR is an established risk marker of CKD but it should be used in conjunction with others when eGFR is not low (<60ml/min/1.73m2). Current data suggested the population included here is not particularly of impaired renal filtration. Relevant discussion on eGFR in comparison to NCR should be shortened and organized.

Response

Thank you for this insight.

Because eGFR is already an established risk marker, we have reduced discussion of it.

Minor comments

To make it less confusing, only keep the number of included subjects in the abstract and delete the total number screened.

Response 

As suggested, the number in the baseline population has been deleted from the abstract; only the number of participants is specified (line 30).

Line 127 of page 6, "an" MCP should be "a" MCP.

Response 

The text has been changed as requested.

In the methods section, please specify the name and version of statistical software.

Response 

The statistical software is now specified: ‘SPSS 11.01J software for Windows (SPSS, Japan, Tokyo, Japan) was used to perform all statistical analyses.’ (lines 190–191).

In the methods section, please specify the regression models used for the evaluation.

Response 

In the Statistical analyses subsection, we now explain why the logistic regression model was chosen for estimating disability: ‘To estimate risk of disability, we used the multiple logistic regression model rather than the Cox proportional hazards model, because data on the exact date on which participants entered the public LTC insurance program had not been collected. Furthermore, this program was implemented from April 2000 and the baseline survey was conducted in 1993; the LTC service was not available between 1993 and March 2000’ (lines 169–173).

We also used the Cox proportional hazards regression model to estimate the risk of all-cause mortality (please see our response to Reviewer #1’s second comment).

Table 4 has duplicated results with Table 3 (the "entire cohort"). Similarly Figure 2 is a graphical presentation of some results presented in Table 3. Such items should be removed.

Response 

We have deleted the results for the entire cohort from the multivariable analysis results presented in Table 4. We have also deleted Figure 2.

Conclusion "public LTC insurance services" should be replaced by disability or disability determined by public LTC insurance services.

Response 

As requested, ‘subsequent need for public LTC insurance services’ has been replaced by ‘disability’.

Reviewer #2: In this manuscript, the author has demonstrated that the activity of urinary N-acetyl-β-glucosaminidase (NAG), a marker of kidney injury, is associated with subsequent risk of disability or early death in a general population. However, there exists several problems.

1、 What is the specific date on which to determine whether the events occurred or not？Please describe the medium and IQR of the follow-up period.

Response 

The follow-up period is now mentioned in the abstract (‘A total of 1182 participants were followed up for a median of 12.4 years.’; line 30) and Participants and Methods (‘the mean and median follow-up periods were 11.4 years and 12.4 years, respectively’; lines 156–157).

2、 The association between the quartile groups of NAG groups and disability or death should be evaluated using the Kaplan-Meier survival method and compared using log-rank statistics.

Response 

Participants were not asked their exact date of entry into the LTC insurance program. Therefore, we were unable to use the Kaplan–Meier survival method to estimate the incidence of disability. Instead, we used the model to estimate all-cause mortality. However, the results were equal to those of the univariate analysis obtained using the Cox proportional hazards model.

If our study had been a randomized controlled trial, it would have been helpful to present the cumulative survival rate in figure form, because all confounders had already been adjusted for at the time of randomization. However, the cohort study results require adjustment for potential confounders; simply presenting the results of univariate analysis in figure form may lead to misunderstanding of the results.

3、 How about the relationships between urinary NAG and other clinical variables?

Response 

Thank you for making this important point. To confirm collinearity, we performed a correlation analysis between urinary NAG and other clinical variables (see Table 3).

4、 The association between the NAG and cardiovascular mortality and renal mortality should better be evaluated.

Response 

We did not set end-stage renal failure as an endpoint. Therefore, we were unable to determine whether a high urinary NAG to creatinine ratio was associated with increased risk of end-stage renal failure.

We have previously used the Cox proportional hazards model to investigate the relation between urinary NAG activity and cardiovascular events. However, we found no association between increased urinary NAG activity and risk of cardiovascular disease.

5、 What is your criteria for selecting independent variables in your multivariable-adjusted model？

Response 

We included traditional cardiovascular risk factors as covariates, because most cardiovascular risk factors are also risk factors for CKD. Furthermore, to adjust for potentially underlying disability and diseases among the participants, we also used physical activity as a covariate (lines 174–183).

6、 The usage of anti-hypertensive drugs, anti-diabetic drug, statins and comorbid conditions should be recorded and adjusted.

Response 

Unfortunately, participants were not asked whether they had used these drugs during the follow-up period. Therefore, we are unable to adjust the analyses for these confounders.

7、 In order to determine whether risk prediction models were improved by addition of the NAG, C-index should be calculated for the demographic, eGFR, and cardiovascular risk factor model for each outcome.

Response 

Thank you for your helpful suggestion. It would be useful if we could create a new model for predicting future disability. However, the aim of this paper was to establish an association between NAG/creatinine ratio and disability or early death. Creation of a new model for predicting future disability is beyond the scope of this paper. However, we will try to do this in the future.

8、 Q1 needs to be placed in the table, and the specific values of Q1-Q4 need to be marked.

Response 

Thank you for your helpful comment. We have made the changes requested.

9、 NAG is one of the urine kidney injury biomarkers, not equals to CKD. These two concepts should not be confused.

Response 

Thank you for this observation.

We regard NAG activity as a kidney injury marker and CKD as a consequence of chronic kidney injury.

10、 In line 277-279, you mentioned that “The association between urinary NAG/creatinine ratio and disability or death, which was found to be independent of cardiovascular risk factors (including age), suggests that CKD is related to the physiological changes of aging rather than those of cardiovascular disease. ”, while I am so confused about this sentence. How can I come to a conclusion that “CKD is related to the physiological changes of aging” from your data? The discussion should be re-arranged.

Response 

We agree that this statement was confusing and an overstatement. We have replaced it with discussion about the mechanisms underlying CKD and aging, with reference to several relevant studies (lines 392–401).

11、 In your opinion, what is the probably reason that urinary NAG/creatinine ratio can not predict subsequent disability or death in patients with proteinuria?

Response

As mentioned in the Limitations subsection, ‘the great majority of the total cohort did not have proteinuria’ (lines 431–432); we considered the number of participants with proteinuria (n=56) to be too low to allow estimation of the risk of disability or death.

12、 I think Table 2 and Figure 2 maybe not necessary.

Response

Thank you for your comment. We have deleted Figure 2 and amended Table 2.

Reviewer #3: The description of the study design in this article is incomplete and lacks specific follow-up methods. There are also doubts about the statistical methods used in this article to illustrate the relationship between urinary NAG/creatinine ratio and the risk of death and disability. So I don't think this article has reached the standard for receiving manuscripts. Questions are as follows:

1. What is the collection method of the outcome event data? At what time were the collection points?

Response 

We mailed the questionnaires to the participants or their relatives at the end of December 2005 and collected outcome data between January and March 2006. For details, please see the subsection The follow-up study of Participants and Methods; lines 153–159).

2. Since the beginning of the study in 2005, what was the median follow-up time for all participants? How many subjects were lost to follow-up?

Response 

The median follow-up time was 12.4 years (see Table 1).

A total of 1438 participants who were aged ≥ 65 years at the initiation of the follow-up study at the end of October 2005 were eligible for this study. Only 45 subjects were lost to follow-up (see Results; line 198). Initially, we collected 1400 responses. However, during the Cox hazards regression analysis, we found several faults in the responses (e.g. no descriptions of the date of death). Therefore, 1393 completed questionnaires were used.

3. The statistical method used in the analysis of survival data in this paper used multivariate logistic regression analysis instead of the Cox risk regression model (Table 2 and Table 3), which ignored the impact of censored data.

Response 

Thank you for your comment.

We did not collect data on the exact date of participants’ entry into the public LTC program. Therefore, we were unable to use the Cox regression model. However, if participants were able to discontinue their use of the service, they may be considered to no longer have a disability.

For data from participants who had died, it is preferable to use the Cox regression model. Therefore, in the revised manuscript, we describe how we used the Cox proportional regression model to estimate the relation between baseline urinary NAG/creatinine ratio and future risk of all-cause mortality.

4. In Table 4, participants with proteinuria were only 56, which could produce an unstable model.

Response 

We agree and have revised the manuscript to present the results for risk in participants without proteinuria only.

---

## [Decision Letter · Decision Letter 1]

21 Jan 2022

PONE-D-21-27646R1Association between urinary N-acetyl-β-glucosaminidase activity–urinary creatinine concentration ratio and risk of disability and all-cause mortatilityPLOS ONE

Dear Dr. Tanaka,

Thank you for submitting your manuscript to PLOS ONE. After careful consideration, we feel that it has merit but does not fully meet PLOS ONE’s publication criteria as it currently stands. Therefore, we invite you to submit a revised version of the manuscript that addresses the points raised during the review process.

One of the Reviewers raised some concerns on the statistical analyses. Please carefully answer these queries and consider these suggestions in your revision.

We look forward to receiving your revised manuscript.

Kind regards,

Yan Li, MD, PhD

Academic Editor

PLOS ONE

Journal Requirements:

Reviewers' comments:

Reviewer's Responses to Questions

**Comments to the Author**

1. If the authors have adequately addressed your comments raised in a previous round of review and you feel that this manuscript is now acceptable for publication, you may indicate that here to bypass the “Comments to the Author” section, enter your conflict of interest statement in the “Confidential to Editor” section, and submit your "Accept" recommendation.

Reviewer #1: All comments have been addressed

Reviewer #2: (No Response)

Reviewer #3: All comments have been addressed

2. Is the manuscript technically sound, and do the data support the conclusions?

Reviewer #1: (No Response)

Reviewer #2: Yes

Reviewer #3: Yes

3. Has the statistical analysis been performed appropriately and rigorously? 

Reviewer #1: (No Response)

Reviewer #2: No

Reviewer #3: Yes

4. Have the authors made all data underlying the findings in their manuscript fully available?

Reviewer #1: (No Response)

Reviewer #2: Yes

Reviewer #3: Yes

5. Is the manuscript presented in an intelligible fashion and written in standard English?

Reviewer #1: (No Response)

Reviewer #2: Yes

Reviewer #3: Yes

6. Review Comments to the Author

Reviewer #1: (No Response)

Reviewer #2: Thanks for your detailed modification, and there exists minor questions as following:

1、 Why chose Kruskal–Wallis non-parametric test for continuous variables, instead of ANOVA? Were all continuous variables skewness distribution?

2、 In Table 1, the P value between any two comparisons should be marked, especially when compared to NA group.

3、 In Table 2, since urinary NAG/creatinine ratio quartile is a ranked variable, P-trend value and the P value for Q2\\Q3\\Q4 compared to Q1 should be listed. Therefore, as for ranked variable, Pearson chi-square test is not suitable for categorical variables to estimate P values.

Reviewer #3: (No Response)

7. PLOS authors have the option to publish the peer review history of their article (what does this mean?). If published, this will include your full peer review and any attached files.

Reviewer #1: No

Reviewer #2: No

Reviewer #3: No

---

## [Author Response · Author response to Decision Letter 1]

24 Feb 2022

Reviewer #2: Thanks for your detailed modification, and there exists minor questions as following:

1、 Why chose Kruskal–Wallis non-parametric test for continuous variables, instead of ANOVA? Were all continuous variables skewness distribution?

Response

Thank you for drawing our attention to this point.

We checked the distribution of all continuous variables and confirmed that the data for several variables follow a normal distribution; these data were then reanalyzed using ANOVA.

We have accordingly changed the sentence on lines 162–164 (page 7) to read, ‘Continuous variables, expressed as means and standard deviations or medians and interquartile ranges, were compared by means of analysis of variance or the Kruskal–Wallis non-parametric test.’

2、 In Table 1, the P value between any two comparisons should be marked, especially when compared to NA group.

Response

Thank you for your comment.

In addition to using ANOVA or the Kruskal–Wallis test to find significant differences among the NA, Disability, and Had died groups, we used the Tukey or Steel–Dwass test to compare data for the NA group versus the Disability group and the NA group versus the Had died group.

This is now stated on lines 165 and 166 (page 7) (‘We also used the Tukey test or Steel–Dwass test to compare data for the continuous variables between the groups.’), and the statistical package is named on lines 193 and 194 (page 8) (‘Additionally, we used R version 4.1.2 (the R Foundation for Statistical Computing, Vienna, Austria) software for the Steel–Dwass test.’). Relevant p values and explanatory footnotes have been added to Table 1.

3、 In Table 2, since urinary NAG/creatinine ratio quartile is a ranked variable, P-trend value and the P value for Q2\\Q3\\Q4 compared to Q1 should be listed. Therefore, as for ranked variable, Pearson chi-square test is not suitable for categorical variables to estimate P values.

Response

Thank you for making this important point.

We used the logistic regression model, instead of the Pearson chi-square test, to estimate differences among quartiles. Furthermore, we calculated p values for Q2, Q3, or Q4 versus Q1; significant differences are highlighted with asterisks in Table 2.

Use of the logistic regression model is now mentioned in the sentence on lines 164 and 165 (page 7) (‘Categorical variables, expressed as proportions, were compared using the chi-square test or the logistic regression model.’) and in Table 2, footnote c.

---

## [Decision Letter · Decision Letter 2]

7 Mar 2022

Association between urinary N-acetyl-β-glucosaminidase activity–urinary creatinine concentration ratio and risk of disability and all-cause mortatility

PONE-D-21-27646R2

Dear Dr. Tanaka,

We’re pleased to inform you that your manuscript has been judged scientifically suitable for publication and will be formally accepted for publication once it meets all outstanding technical requirements.

Kind regards,

Yan Li, MD, PhD

Academic Editor

PLOS ONE

Additional Editor Comments (optional):

Reviewers' comments:

Reviewer's Responses to Questions

**Comments to the Author**

1. If the authors have adequately addressed your comments raised in a previous round of review and you feel that this manuscript is now acceptable for publication, you may indicate that here to bypass the “Comments to the Author” section, enter your conflict of interest statement in the “Confidential to Editor” section, and submit your "Accept" recommendation.

Reviewer #1: (No Response)

Reviewer #2: All comments have been addressed

Reviewer #3: All comments have been addressed

2. Is the manuscript technically sound, and do the data support the conclusions?

Reviewer #1: (No Response)

Reviewer #2: Yes

Reviewer #3: Yes

3. Has the statistical analysis been performed appropriately and rigorously? 

Reviewer #1: (No Response)

Reviewer #2: Yes

Reviewer #3: Yes

4. Have the authors made all data underlying the findings in their manuscript fully available?

Reviewer #1: (No Response)

Reviewer #2: Yes

Reviewer #3: Yes

5. Is the manuscript presented in an intelligible fashion and written in standard English?

Reviewer #1: (No Response)

Reviewer #2: Yes

Reviewer #3: Yes

6. Review Comments to the Author

Reviewer #1: (No Response)

Reviewer #2: (No Response)

Reviewer #3: (No Response)

7. PLOS authors have the option to publish the peer review history of their article (what does this mean?). If published, this will include your full peer review and any attached files.

Reviewer #1: No

Reviewer #2: No

Reviewer #3: No

---

## [Editor Report · Acceptance letter]

17 Mar 2022

PONE-D-21-27646R2 

Association between urinary *N*-acetyl-β-glucosaminidase activity–urinary creatinine concentration ratio and risk of disability and all-cause mortality 

Dear Dr. Tanaka:

I'm pleased to inform you that your manuscript has been deemed suitable for publication in PLOS ONE. Congratulations! Your manuscript is now with our production department. 

Kind regards, 

on behalf of

Professor Yan Li 

Academic Editor

PLOS ONE